# Healing Together: A Narrative Review on How Psychiatric Treatment for Parental Depression Impacts Children

**DOI:** 10.3390/ijerph21030367

**Published:** 2024-03-19

**Authors:** Michelle Cross, Yasmeen Abdul-Karim, Amy Johnson, Colleen Victor, Andrew Rosenfeld

**Affiliations:** Larner College of Medicine at the University of Vermont, Psychiatry Department, Vermont Center for Children, Youth & Families, Burlington, VT 05401, USA

**Keywords:** family-based care, depression, major depressive disorder, therapy, psychotherapy, child psychiatry, child mental health, psychopharmacology, medication treatment, parent therapy, parenting skills training

## Abstract

It is well known that parental depression is correlated to adverse child mental health outcomes; but what is the effect of treating parental depression on the child? This narrative review aims to explore this question, and how certain specific interventions designed to help depressed parents affect mental health outcomes in their children. The academic database APA PsychInfo was searched for articles that broadly included interventions for parents with depression as well as child wellbeing or outcomes as of October 2023. Additional searches were conducted in the academic database PubMed in December 2023 and January 2024. Forty-nine articles met the inclusion criteria and were examined closely for this review. The studies included were divided into the following categories: psychotherapy, psychopharmacology, parenting support, and paternal interventions. We discuss the implications of our review on clinical practice and recommend further research in this area.

## 1. Introduction

We are in the midst of a national mental health crisis. One in five adults currently live with a mental illness [1], and one in five youth aged 13–18 currently or at some point in their life have had a seriously debilitating mental illness [2]. In other words, many families are struggling. In order to address this crisis and develop effective interventions, it is, therefore, necessary to gain a deeper appreciation and understanding of the family dynamic factors that may contribute to the etiology of psychiatric illness and wellness.

Research has consistently shown a correlation between parental mental health and behavioral and emotional problems in children. This relationship between a family member with mental illness and the long-term health outcomes on the child is so strong that this is considered an adverse childhood experience which contributes to lifelong adverse health outcomes in offspring [3]. Children with depressed mothers are more likely to exhibit internalizing behaviors [4], externalizing behaviors, anxiety, and depression [5]. Furthermore, in the children of parents with depression, there is a significantly increased risk of the development of depression, anxiety, and substance use disorders during adolescence and young adulthood [6]. Concerningly, one study of hospitalized adolescents found there was a two-fold increased risk of attempted suicide in adolescents who had at least one parent with a mental health concern [7].

Aside from biological contributors, which by some estimates from twin studies account for around 40% of variation in the major depressive disorder phenotype [8], research has also begun to elucidate how parental depression may be influencing environmental and psychological contributors to depression in offspring. The key to this appears to be in the interaction and relationship between child and parent. Previous research indicates that depressed mothers are more likely to display more punitive parenting practices, engage in less positive parenting, be less responsive, and have a more negative affect towards their children [9]. In addition, childhood exposure to early-life stress associated with parental depression, negative parenting practices, inconsistent parenting practices, and modeling of depressed cognitions and behaviors are thought to contribute to the increased risk of transmission of depression to offspring [5].

As childhood mental health and wellness are inextricably linked to the mental health of the children’s parents, it appears that part of the solution to the childhood mental health crisis will require a multidimensional family-based approach. In other words, in order to help children, we must also help their parents. This review aims to assist clinicians in the treatment of the family by discussing evidence-based interventions for depressed parents that may be utilized to effectively improve childhood outcomes.

## 2. Materials and Methods

This review was conducted to determine if we could find a relationship between the treatment of parental depression and offspring wellbeing. If this relationship was found, we then aimed to explore the effects of different parent-centered interventions on children.

### 2.1. Eligibility Criteria

A search was conducted for studies that included both treatments of depression in parents and subsequent childhood outcomes. Treatments of depression in parents were narrowed down to categories of evidence-based psychotherapy, psychopharmacology, and parenting interventions. There was no restriction on publication date or country. Studies were, however, restricted to those written in English. Articles were excluded if they were not research studies. Studies of maternal depression which did not include any mention of childhood outcomes were largely excluded. Due to the low number of paternal studies relative to maternal studies on this topic, paternal studies that excluded childhood outcomes were included.

### 2.2. Data Sources and Search Strategy

A search strategy was developed with assistance from a university librarian (AD) who has extensive experience undertaking literature searches within the mental health and medical fields (Appendix A).

The preliminary search was conducted via the APA PsychInfo database. The search broadly included interventions for parents with depression as well as child wellbeing or outcomes by utilizing both keywords and Medical Subject Headings (MESH) terms. This search was conducted in October 2023. Additional searching was conducted in PubMed by the authors to provide additional support and background of studies found in the initial search between December 2023 and January 2024. This resulted in 70 total articles.

### 2.3. Screening

All authors were assigned 10–15 articles to screen for eligibility and appropriateness for inclusion in this review. These articles were then discussed as a team and categories for organizing articles were developed collaboratively. Ultimately, 20 articles were excluded and 49 articles were included in the final dataset.

### 2.4. Data Summary and Synthesis

Each author was assigned to a category developed as a team in the data screening stage. These categories included Abstract/Introduction/Methods/Discussion, Psychotherapy, Psychopharmacology, Paternal Interventions, and Parenting Interventions. Each author was responsible for a thorough review and summary of articles within their assigned topic.

## 3. Results

### 3.1. Psychotherapy

When investigating the family sequelae of maternal depression, some studies evaluated whether children’s emotional and/or behavioral states improve following treatment of maternal depression through psychotherapy interventions (Table 1). In general, these studies only assessed maternal depression rather than assessing other adults in the home, though many articles pointed out the potential relationship between the mental health of other adults and children in the household. Most studies only included children without significant developmental delays, and some included only children without psychiatric diagnoses. The majority of studies only assessed biological children, and all required that the mother in the study be the primary caretaker for the child. The specific therapeutic modalities primarily investigated, and thus reviewed here, include cognitive behavioral therapy (CBT), interpersonal therapy (IPT), and child–parent psychotherapy (CPP).

A 2015 meta-analysis combined RCTs evaluating therapy for depressed mothers, either perinatal or otherwise [10]. This showed medium effect sizes for both maternal (g = 0.64) and child (g = 0.4) outcomes based on the psychological treatment of the mothers’ depression. Medium effect sizes were also found for an improvement in the mother–child relationship (g = 0.4) and a reduction in parenting/marital distress (g = 0.77). The authors’ hypothesis was that improvements in parenting warmth and sensitivity are the mediating factors connecting successful maternal depression treatment with improved child outcomes.

Attachment-focused interventions leverage the power of caregiver–infant bonds to influence emotional regulation capacities. A longitudinal study [11] found that children of mothers with depression who received CPP had significantly higher rates of secure attachment compared with both children of mothers with depression who did not receive intervention and children of mothers without depression. These results suggest that maternal depression in early childhood may impact cohesive mother–child interactions even more strongly than attachment patterns. Another program utilized CPP over 9 months with mother–child dyads to assess benefits for preventing depression and PTSD symptoms [12]. Not only did depression and PTSD symptoms decrease significantly for mothers across the course of the study, but children experienced similar significant improvements.

The approach of CBT interventions includes identifying and adjusting ineffective beliefs and behaviors. This applies to maternal depressive illness based on observations that depressed moms can be less responsive to their infants, experience more parenting stress, and rate their infants’ temperaments as more negative [13]. It is thus likely that mothers’ depressive symptoms create a negative bias affecting their perceptions of their children’s behavior. Other studies that utilized maternal reports of children’s behavior also noted the negative reporting bias of depression as a confounding factor. CBT treatment aims to alter this negative bias and subsequent related behaviors in order to influence children’s mental health outcomes.

One CBT study [14] assessed mothers’ negative causal attributions of their children’s misbehaviors in dyads including children with ADHD. Adding CBT to behavioral parent training led to a significant increase in positive child-crediting attributions, though without significant changes to child-blaming attributions or maternal expectations regarding child compliance. A 2017 systematic review of CBT studies [11] showed some advantages for antenatal depression, including for infant outcomes such as self-regulation and reactivity as well as maternal outcomes such as attentional bias. A “very well-designed” [9] RCT with a low risk of bias was included and found improvements across a host of infant outcomes at 9-month follow-up, including several large effect sizes. When this cohort was followed at 2 years post-intervention [15], measures of total parenting stress were significantly reduced. This included child-centered scores such as the adaptability scale measuring self-regulation. These effects were not maintained at the 5-year follow-up mark, perhaps because the control group had shown significant improvements in affective symptoms by that time and also had a higher proportion of mothers taking medications compared to the CBT intervention group [16]. For postpartum depression, CBT did not stand out compared to other therapy approaches, though multiple studies showed parenting stress decreasing after CBT interventions.

Another study [9] focused on a marginalized population of mothers with depression from low-income families and predominantly from minority racial–ethnic backgrounds. Engagement in the treatment was low, with less than a third of participants receiving enough weeks of medication or CBT to qualify as an adequate “dose”. However, amongst the mothers assigned to active treatment, their children’s behavioral problems and adaptive skills showed a trend of improvement across the 12 months of follow-up. Moreover, amongst the minority of mothers whose depression was remitted, their children showed statistically significant improvement in behavioral problems. This research underscores the need for community-centered outreach and treatment innovation to both engage and treat families affected by systemic inequities. Given the effectiveness of CBT for treating maternal depression itself and the challenges of service delivery, Loughnan [17] created a three-session standalone Internet CBT program for postpartum depression. Not only did this show large effect sizes for the treatment of both maternal depression and anxiety but it also improved maternal bonding (g = 0.70) alongside parental confidence.

Alternatively, some articles focused on the relational aspects of mothering as a potential factor contributing to child outcomes by using interpersonal psychotherapy (IPT) approaches. A systematic review [11] showed positive but non-significant results of IPT effects on mother–child relationship quality. The review also covered research demonstrating improvements in verbal intelligence and eventually in attachment security after IPT for depressed mothers, with a separate study supporting improved parent–child relationship quality post-intervention. Though IPT did not separate from a comparator mother–child psychotherapy in another trial, both interventions showed significant improvements compared to the control group in terms of child adaptability, parent–child relationship markers, and positive maternal affect and verbalization. While a group IPT treatment showed enhanced positive maternal perceptions of attachment, these effects did not last over several years of follow-up. Despite these promising results, there were also two studies of perinatal treatment that showed some reduction in parenting stress but lacked group differences on many other measures.

In contrast, another study [18] suggested that relationship-building and engagement strategies mitigated depressive symptoms, perhaps through perceived self-efficacy and social support. This occurred no matter whether these skills were developed through IPT versus supportive treatment. One innovative feasibility study [19] addressed the finding that depressed moms of psychiatrically ill children had higher cardiovascular child-focused stress reactivity when compared with non-depressed mothers of psychiatrically healthy children. IPT-MOMS was provided to depressed moms of psychiatrically ill children as a nine-session psychosocial intervention based on principles of interpersonal psychotherapy. This resulted in depressed mothers’ cardiovascular reactivity moving towards that of non-depressed mothers.

It is possible that studies finding no appreciable effects of IPT on child outcomes may not have been measured at appropriate timelines. Another study utilizing IPT-MOMS [20] concluded that the treatment of maternal depression with evidence-based psychotherapy in families where children also experience internalizing disorders is likely to result in the rapid improvement of maternal symptoms, but one can expect a 6-month delay in positive impacts on offspring as mothers make changes in their parenting behaviors after their depression resolves. However, it is notable that the positive effects of improved maternal depression on child outcomes were not apparent in mothers who reported a history of early sexual abuse or neglect.

**Table 1 ijerph-21-00367-t001:** Effect of Therapy for Depression in Mothers on Children.

	Intervention	Type of Study	Inclusion Criteria	Sample Size	Outcome Measures	Summary of Major Results
Letourneau N. et al., 2017 [11]	IPT, CBT, peer support, maternal–child interaction guidance, CPP, infant massage, non-directive counseling combined with antidepressant medication.	Systematic review	- Women experiencing mild–severe depression in the antenatal or postpartum period. - Inclusion of parenting and/or child development and health outcomes. - Studies included were either randomized controlled trials or quasi-experimental.	>5000 women	Child development, parental behaviors, maternal–child interaction/attachment.	CPP - Children of mothers with depression who received CPP had significantly higher rates of secure attachment compared with both children of mothers with depression who did not receive intervention and children of mothers without depression. CBT - CBT delivered to the mother in the antenatal period showed improvements in infant outcomes in self-regulation and reactivity as well as maternal outcomes in attentional bias. IPT - Improvements in verbal intelligence and attachment security; - Significant improvements compared to the control group in terms of child adaptability, parent–child relationship markers, and positive maternal affect and verbalization.
Aschbacher et al., 2022 [12]	Child–parent psychotherapy (CPP).	RCT	Biological mothers, aged 18 and over, with children between 2 and 6 years of age, who had been exposed to interpersonal trauma, and were fluent in English and/or Spanish.	43 mother–child dyads	Maternal: - Post-traumatic stress scale interview; - Center for Epidemiologic Studies Depression Scale Revised (CESD-R). Child: - Depression and post-traumatic stress symptom subscales of the Trauma Symptom Checklist for Young Children (TSCYC).	Decreased depression (*p* < 0.001, d = −0.84) and PTSD (*p* < 0.001, d = −0.83) symptoms in mothers. Decreased depression (*p* < 0.01, d = −0.53) and PTSD (*p* < 0.01, d = −0.51) symptoms in children.
Cuijpers et al., 2015 [10]	Psychotherapy for depressed mothers (CBT, IPT, counseling, psychodynamic therapy).	Meta-analysis of RCTs	- Depressed mothers or pregnant women whose children were <18 years old.	553	- Mental health of the children; - Quality of interaction between mother and child; - Parenting/marital distress.	- Medium effect sizes for both maternal (g = 0.64) and child (g = 0.4) outcomes based on psychological treatment of the mothers’ depression. - Medium effect sizes were found for improvement in the mother–child relationship (g = 0.4) and reducing parenting/marital distress (g = 0.77).
Novick et al., 2022 [14]	Cognitive behavioral therapy added to behavioral parent training.	Q	- Children with ADHD and their mothers with at least a mild level of depressive symptoms - Children 6–12 years with an IQ of at least 70 by the WISC-IV	98	Maternal: - Structured Clinical Interview for DSM–IV (SCID). Child: - Schedule for Affective Disorders for School-Aged Children Version 4 (KSADS); - Child IQ screen. Family: - Parent–child interaction observational protocol.	Adding CBT to behavioral parent training led to a significant increase in positive child-crediting attributions (*p* = 0.01), though without significant changes to child-blaming attributions or maternal expectations regarding child compliance.
Milgrom et al., 2019 [16]	CBT (five years post-intervention).	RCT follow-up	Women < 30 weeks pregnant with a depressive disorder at recruitment.	25 women	Maternal: - Beck Depression Inventory-II; - Beck Anxiety Inventory; - Parenting Stress Index (PSI). Child: - Bayley Scales of Infant Development (BSID-III); - Child Behavioral Checklist (CBCL), parent report; - Wechsler Preschool and Primary Scale of Intelligence (WPPSI-III).	At 2 years post-intervention: - Reduced total parenting stress (d = 1.47, *p* < 0.05); - Improved child adaptability (d = 0.88, *p* < 0.05). At 5 years post-intervention: - Lower proportion of psychotropic medication in children of those in the CBT intervention group.
Coiro et al., 2012 [9]	CBT and medication.	RCT	Low-income women with major depression and their children ages 4 to 11.	60 mother–child dyads	Maternal: - Hamilton Depression Rating Scale. Child: - Behavior Assessment System for Children (parent report).	- Low treatment engagement; less than ⅓ received enough weeks of medication or CBT to qualify as an adequate “dose.” - Amongst the mothers assigned to active treatment, their children’s behavioral problems and adaptive skills showed a trend of improvement across the 12 months of follow-up. - In mothers in which depression remitted, their children showed statistically significant improvement in behavior problems
Loughnan S. et al., 2019 [17]	3-session standalone Internet CBT program for postpartum depression.	RCT	Women >18 within 12 months postpartum, meeting clinical criteria for anxiety and/or depression.	79 total participants	Maternal: - GAD-7; - PHQ-9; - EPDS; - Kessler 10-item Psychological Distress scale; - Maternal Postnatal Attachment Scale; - Karitane Parenting Confidence Scale; - World Health Organization Quality of Life scale.	- Large effect sizes for treatment of both maternal depression and anxiety, and improved maternal bonding (g = 0.70) alongside parental confidence.
Beeber et al., 2013 [18]	IPT+ depression-specific parenting enhancement.	RCT	Mothers and their infants enrolled in an Early Head Start program.	226 mother–infant dyads	Maternal: - Hamilton Rating Scale for Depression; - Structured Clinical Interview for DSM–IV (SCID); - MOS Health Form; - Everyday Stressors Index; - Interpersonal Inventory; - General Self-Efficacy Scale; - Social Support Seeking Inventory. Family: - Maternal–Child Observation (MCO); - HOME Inventory; - Family Information Interview.	Relationship-building and engagement strategies, no matter whether these were developed through IPT versus supportive treatment, mitigated depressive symptoms in mothers (*p* < 0.0001).
Swartz et al., 2018 [20]	IPT-MOMS.	RCT	Mothers with depression aged 18–65 of children with at least one internalizing disorder aged 7–18.	62 mother–child dyads	Maternal: - Structured Clinical Interview for DSM-IV Axis I Disorders (SCID-I); - Personality Disorders (SCID-II) - Clinician-rated HRSD-25; - Work and Social Adjustment Scale (WSAS); - Childhood Trauma Questionnaire (CTQ). Children: - Kiddie Schedule for Affective Disorders and Schizophrenia—Present and Lifetime Version (K-SADS-PL); - Child Depression Inventory (CDI); - Strengths and Difficulties Questionnaire (SDQ); - Columbia Impairment Scale (CIS); - If the child was <age 11, the mother completed the SDQ about the child. Family: - Child Report of Parent Behavior Inventory (CRPBI).	- Mothers in these families rapidly responded to treatment, with a 6-month delay in impact on their children’s internalizing symptoms. - Improved maternal depression appeared to have little to no impact on childhood outcomes in mothers who reported a history of early childhood sexual abuse or neglect.

While heterogenous methodologies and timelines hamper easy summarization, there appears to be abundant evidence for the general benefits of providing psychotherapy treatments to mothers with depression. These benefits extend to infants and children, though age and socio-demographic mediators have received minimal attention. Some work suggests that mediators of positive childhood outcomes include parenting behaviors that affect attachment and relationship quality, but this connection remains to be replicated and clarified. Also less clear is which therapy modality is favored, due to the lack of head-to-head comparisons in relevant populations. Finally, specifying the mechanism of action for each beneficial effect of parent-targeted therapeutic interventions remains challenging with the existing literature.

### 3.2. Psychopharmacology

Amongst the articles that explored the psychopharmacological treatment of the parent, specifically the mother, overall results showed that the treatment of maternal depression led to improvement in children’s symptomatology and functioning (Table 2).

Nonpsychotic depressed mothers treated with antidepressant medication were evaluated as part of the Sequenced Treatment Alternatives to Relieve Depression (STAR*D) study to understand what response this had on the psychosocial outcomes of their children [21,22,23]. Mothers’ initial diagnoses were established by clinical interview and confirmed using a symptom checklist based on the Diagnostic and Statistical Manual of Mental Disorders, Fourth Edition (DSM-IV). Depression severity was measured using the Hamilton Rating Scale for Depression (HRSD), with remission defined as a score of 7 or less and response defined as a 50% or greater reduction of the baseline HRSD score. The children were screened diagnostically via the use of selected sections from the Kiddie Schedule for Affective Disorders and Schizophrenia for School-Age Children—Present and Lifetime Version. Additionally, child psychopathology, including internalizing and externalizing symptoms, overall adjustment, and family/parental functioning, was assessed. Symptoms were collected at baseline and 3-month intervals.

A total of 151 mother–child pairs from the STAR*D trial participated in the study with 114 remaining in the study at the time of the 3-month assessment. Mothers’ ages ranged from 25 to 60 years old. Children were aged 7 to 17 years and at the initiation of maternal treatment, approximately one-third had a current psychiatric disorder. Mothers who dropped out were not significantly different at baseline from those who remained in the study, though mother–child pairs with male children were more likely to drop out than pairs with female children (70% vs. 30%, *p* = 0.01). Of those who received follow-up assessments, 33% of mothers experienced remission of depressive symptoms before the 3-month follow-up with an average time to remission of 55 days. Of those mothers who did not remit by 3 months, they were found to be more depressed at baseline, of lower socioeconomic status, and have comorbid anxiety. The remission rate of maternal depression after 3 months of medication treatment was significantly associated with reductions in children’s diagnoses, as well as internalizing, externalizing, and total symptoms [23]. Additionally, changes in maternal expressions of warmth and acceptance were associated with changes in children’s internalizing symptoms [21]. The investigators found not only that the children of mothers with early remission (by 3 months) experienced favorable improvements, but also that the children of mothers with later remission experienced improved functioning and decreased symptoms [22].

A subsequent study [24] looked at the above cohort and compared the symptoms and outcomes of children one year after maternal remission from depression. These were divided into early remitters (within 0–3 months), late remitters (3–12 months), and non-remitters. During the year that followed remission, the children of mothers with remission (early or late) were found to have decreased overall psychiatric symptoms and internalizing behaviors. Of note, child externalizing behaviors increased for non-remitting mothers. Additionally, children’s overall functioning was improved only in those with mothers who had early remission.

**Table 2 ijerph-21-00367-t002:** Effect of Medication for Depression in Mothers on Children.

	Intervention	Type of Study	Inclusion Criteria	Sample Size	Outcome Measures	Summary of Major Results
Weissman et al., 2006 [6]	Antidepressant treatment of depressed mothers.	RCT (part of the Sequenced Treatment Alternatives to Relieve Depression, or STAR*D trial).	Nonpsychotic depressed mothers aged 25–60 years with at least one child aged 7–17 in the home at least 50% of the time.	151 mother–child dyads	Maternal: - Clinical interview and symptom checklist based on DSM-IV criteria for major depressive disorder; - Hamilton Rating Scale for Depression; - Quick Inventory of Depressive Symptomatology Self-Report. Child: - Structured interview with mom and children directly using the Kiddie Schedule for Affective Disorders and Schizophrenia—Present and Lifetime Version; - Child Behavioral Checklist, Parent Version; - Clinician–Child Global Assessment Scale.	The remission rate of maternal depression after three months of medication treatment was significantly associated with the following: - Reductions in the diagnoses of internalizing disorders in their children (*p* = 0.03), but not externalizing disorders; - Larger reductions in their children’s internalizing (*p* < 0.001), externalizing (*p* = 0.004), and total (*p* < 0.001) symptoms as compared to children of mothers who did not remit.
Foster et al., 2008 [21]	Antidepressant treatment of depressed mothers.	RCT (part of the Sequenced Treatment Alternatives to Relieve Depression, or STAR*D trial).	Nonpsychotic depressed mothers aged 25–60 years with at least one child aged 7–17 in the home at least 50% of the time.	151 mother–child dyads	Maternal: - Clinical interview and symptom checklist based on DSM-IV criteria for major depressive disorder; - Hamilton Rating Scale for Depression; - Quick Inventory of Depressive Symptomatology Self-Report. Child: - Child Behavior Checklist, Parent Version; - Social Adjustment Inventory for Children and Adolescents (child as informant). Family: - Children’s Report of Parenting Behavior Inventory; - Family Relationship Index.	Increased maternal expression of warmth and acceptance were associated with a decrease in children’s internalizing symptoms (*p* = 0.000).
Pilowsky et al., 2008 [22]	Antidepressant treatment of depressed mothers.	RCT (part of the Sequenced Treatment Alternatives to Relieve Depression, or STAR*D trial).	Nonpsychotic depressed mothers aged 25–60 years with at least one child aged 7–17 in the home at least 50% of the time.	151 mother–child dyads	Maternal: - Clinical interview and symptom checklist based on DSM IV criteria for major depressive disorder; - Hamilton Rating Scale for Depression; - Quick Inventory of Depressive Symptomatology Self-Report. Child: - Structured interview with mom and children directly using the Kiddie Schedule for Affective Disorders and Schizophrenia—Present and Lifetime Version; - Child Behavioral Checklist, Parent Version; - Clinician–Child Global Assessment Scale.	Remission of mother’s depression (both early and late) is associated with improved functioning (early remitters, *p* = 0.006; late remitters, *p* = 0.001) and decreased psychiatric symptoms in their children (child-reported symptoms: early remitters, *p* = 0.0001; late remitters, *p* = 0.03. Mother-reported child symptoms: early remitters, *p* = 0.02; late remitters, *p* = 0.0005).
Wickramaratne et al., 2011 [24]	Antidepressant treatment of depressed mothers.	RCT (part of the Sequenced Treatment Alternatives to Relieve Depression, or STAR*D trial).	Nonpsychotic depressed mothers aged 25–60 years with at least one child aged 7–17 in the home at least 50% of the time.	151 mother–child dyads	Maternal: - Clinical interview and symptom checklist based on DSM-IV criteria for major depressive disorder; - Hamilton Rating Scale for Depression; - Quick Inventory of Depressive Symptomatology Self-Report. Child: - Structured interview with mom and children directly using the Kiddie Schedule for Affective Disorders and Schizophrenia—Present and Lifetime Version; - Child Behavioral Checklist, Parent Version; - Clinician–Child Global Assessment Scale.	During the year that followed remission, the children of mothers with remission (early or late), were found to have the following: - Decreased overall psychiatric symptoms (child-reported symptoms: early remitters, *p* < 0.01; late remitters, *p* < 0.05. Mother-reported child symptoms: early remitters, *p* < 0.05; late remitters, *p* < 0.05) and internalizing behaviors (early remitters, *p* = 0.03; late remitters, *p* = 0.051). Child externalizing behaviors worsened for non-remitting mothers (*p* < 0.05). Children’s overall functioning was significantly improved only in those whose mothers had early remission (*p* < 0.01).
Weissman et al., 2014 [25]	Antidepressant treatment of depressed mothers.	RCT, followed by an open trial.	Nonpsychotic depressed mothers aged 18–65 years and children 7–17 years.	76 treated mothers and their 135 children	Maternal: - Hamilton Depression Rating Scale; - Social Adjustment Scale (SAS); - Parental Bonding Instrument (PBI). Child: - Children’s - Depression Inventory (CDI); - Columbia Impairment Scale (CIS); - Multidimensional Anxiety Scale for Children; - Children’s Global Assessment Scale.	Remission of maternal depression was associated with a decrease in their children’s depressive symptoms on the CDI (*p* < 0.0001) and a relapse of depression in mothers was associated with an increase (*p* = 0.11). Maternal remission was significantly associated with improvement between mother and child related to the following: - Parental functioning (*p* < 0.0001), - Communication—specifically, “able to listen to and talk to my child” SAS item no 2 (*p* < 0.0001), and - Bonding—specifically, PBI overprotection (child report: *p* < 0.01; mother report: *p* < 0.0001).
Weissman et al., 2015 [26]	Antidepressant treatment of depressed mothers.	RCT, followed by an open trial.	Nonpsychotic depressed mothers aged 18–65 years and children 7–17 years.	76 treated mothers and their 135 children	Maternal: - Hamilton Depression Rating Scale; - Social Adjustment Scale; - Parental Bonding Instrument. Child: - Children’s - Depression Inventory; - Columbia Impairment Scale; - Multidimensional Anxiety Scale for Children; - Children’s Global Assessment Scale.	There were statistically significant differences in child outcomes based on medication choices in this study; however, due to the scope of this review, those findings are not shared here. However, readers are encouraged to review the original article as changes in children’s symptomatology and functioning did differ based on antidepressant choice; thus, these findings may warrant further clinical consideration when implementing best practices in supporting mothers to remission.

Another study [25] looked at depressed mothers who participated in a randomized controlled trial of antidepressant treatment for 12 weeks, followed by an open trial for a total of 9 months. Mothers were aged 18–65 years and children 7–17 years. Both had baseline measures and valid assessment instruments conducted: mothers were assessed for depression, adjustment, and parental bonding; children were assessed for depression, impairment, anxiety, and overall functioning. Findings showed that remission of maternal depression was associated with a decrease in their children’s depressive symptoms and that relapse was associated with an increase. Maternal remission was also associated with improved parenting demonstrated by a mother’s ability to listen and talk to her child and improved parental bonding. Additionally, the impact on their children’s symptomatology and functioning did differ based on antidepressant choice and may warrant clinical consideration when implementing best practices in supporting mothers to remission [26].

In conclusion, though the evidence is limited by the small number of studies, there is a robust indication that medication treatment to reduce mothers’ depressive symptoms has positive benefits for their offspring. To the extent there is a signal of the mechanism, it seems to be mediated by relational behaviors associated with positive parenting. This aligns with what has been demonstrated via psychotherapy interventions, as above. The benefits of medication treatment seem to be maintained over time and across several broad domains and measures. However, gaps exist in comparing psychiatric medication treatments to one another, as well as to neurostimulation and/or psychotherapy in head-to-head trials in this domain of parent-focused treatments. Moreover, much of the work to date warrants replication by additional research groups.

### 3.3. What about Dad? Paternal Mental Health and Childhood Outcomes

Fathers’ roles in families manifest diversely, from serving as a primary parental figure to being uninvolved. Families may also have more than one father or father figure, with diverse roles amongst them—including same-sex male couples, step-parents, adoptive and foster parents, and sperm donors. Most studies in this domain focus on biological fathers who are currently involved in caregiving. Though the call for enhanced study of paternal interventions is growing, there exist negligible data on child-focused endpoints after father-focused interventions (Table 3). Here, we will thus focus on the connection between paternal depression and childhood outcomes, in keeping with the emphasis of this review, as well as the focus of what literature exists, on the relationship between paternal mental health and childhood outcomes. 

According to epidemiological studies, fathers develop significant depressive symptoms in the perinatal period at rates approaching those of mothers [27]. Rates are particularly high during the 3 to 6 months postpartum, estimated at around 25% [28]. Paternal peripartum depression rates are notably higher if the mother is also experiencing depression. In turn, children of fathers with depression are at higher risk of excessive crying [29], higher emotion dysregulation [30], and internalizing and externalizing problems [31]. This translates to increased psychopathology in preschool years and increased psychiatric diagnoses by school age [30].

Child effects of paternal perinatal depression may be mediated via changes in parenting behaviors including increased use of corporal punishment, increased parental conflict, and changes to the quality and quantity of father–child interactions [27]. Conversely, the involvement of a non-depressed father may protect against some of the offspring risks associated with maternal depression [32]. Relatedly, paternal depression puts the mother at increased risk for depression as well, indirectly contributing to a further increased childhood risk for psychopathology [33].

A 2020 meta-analysis [34] found converging evidence of the relationship between paternal depressive psychopathology and adolescent depression. This included large and prospective longitudinal studies, suggesting the timing of effects is consistent with a causal relationship. The relationship was less clear when it came to the effects of other varied psychopathologies—paternal substance use, attention-deficit hyperactivity disorder, or post-traumatic stress disorder—on adolescent mood and anxiety. The mechanism of the effects of paternal depression was unclear, with some indication that fathers who are more rejecting or provide less supervision are more prone to having depressed adolescents.

A related systematic review [27] demonstrated childhood internalizing symptoms in the vast majority of qualifying studies assessing the effects of paternal depression. This occurred across childhood age groups and across paternal depressive onset timing, i.e., whether the onset of paternal depression was perinatal or during adolescence. Numerous studies also linked paternal depression with externalizing symptoms in children, such as attention and conduct problems.

While the link between paternal wellbeing and childhood outcomes seems consistent, the mechanism of the connection seems varied. Wang [35] explored risk factors for paternal postpartum depression in a systematic review and meta-analysis. They found converging evidence for paternal factors adding to the risk for depression, including unemployment, neurotic personality traits, mental illness diagnosis, financial strain, pregnancy-/birth-related distress, fewer social supports, avoidant coping, poor self-esteem, negative life events, poor sleep, and perceived stress. The most significant maternal factor influencing paternal peripartum depression was maternal depression itself. Many studies confirmed marital dissatisfaction as a risk factor for paternal perinatal depression. Some evidence showed infant factors, including difficult temperament and difficulty with feeding, contribute to the risk for paternal depression but there were fewer studies in this domain.

These findings provide many opportunities for intervention with depressed fathers. Given the success of home visiting programs for supporting mothers in the perinatal period, Hamil and colleagues [36] devised a Fathers and Babies (FAB) intervention to address their partners. The program is designed with the same cognitive behavioral therapy and attachment principles employed in the highly effective Mothers and Babies (MB) program for maternal postpartum depression prevention, as well as collected input from fathers, mothers, and home-visiting clinicians. The 12-session intervention included digital features and ran concurrently with the MB intervention in families, with a dual focus on how fathers can promote both their own and their partner’s mental health. The pilot program was very well received, feasible, and acceptable, and showed evidence of effectiveness based on the initial focus groups and survey responses.

With a similar line of thinking, Kavanagh et al. [37] created a digital intervention for new mothers and fathers predicated on the enrollment of both parents. The app included modules on caring for babies, role changes, self-care, and supporting one’s partner. While satisfaction with the app was high, only about 37% of participants logged in two or more times and they generally viewed one to three of the nine available modules. While depression and quality of life ratings did not differ significantly by intervention status, parenting self-efficacy scores did improve more with the intervention group. Relationship satisfaction declined across all participants, but less so for recipients of the intervention.

While cognitive behavior therapy interventions have been well studied as highly effective treatments for depression, and there is even some attention paid to CBT as an effective treatment specifically for men, there is little evidence specific to paternal perinatal depression treatment [38]. Attempts to alleviate barriers to paternal involvement in the prevention and treatment of depression leverage technology, group formats, and questioning assumptions about masculinity that might affect help-seeking.

**Table 3 ijerph-21-00367-t003:** Effect of Depression in Fathers on Children.

	Intervention	Type of Study	Inclusion Criteria	Sample Size	Outcome Measures	Summary of Major Results
Wang et al., 2021 [35]	N/A	Systematic review and meta-analysis.	Original observational studies, in English with quantitative measures.	37 studies in systematic review, 17 studies in meta-analysis	Multiple measures, but the Edinburgh Postnatal Depression Screening (EPDS) rating being the most common by far used to assay the association between studied variables and paternal postpartum depressive symptoms.	Six paternal factors were statistically significantly associated with paternal postpartum depression, as well as three maternal or family factors.
Wickersham et al., 2020 [34]	N/A	Systematic review.	English studies since 2000 assaying the relationship between paternal depression and adolescent depression or anxiety with validated measures.	14 studies	Variety of validated measures.	Paternal depression showed the strongest and most consistent correlation with adolescents’ internalizing symptoms, particularly adolescent depression.
Fisher et al., 2012 [33]	N/A	Validation.	Mothers and fathers, cohabitating, shared custody, >18 years old, recruited during early postpartum.	199 couples	- Edinburgh Postnatal Depression Screening (EPDS-P); - Inventory to Diagnose Depression (IDD).	EPDS-P reliable and valid measure of paternal depression.
Kane and Garber, 2009 [31]	N/A	Regression analysis of high-risk family sample.	Fathers, largely White and middle-class.	81 fathers (68 biological, 13 step) and their children	Paternal: - Beck Depression Inventory. Child: - Child Behavior Checklist. Family: - Conflict Behavior Questionnaire.	Paternal depression correlated with child externalizing symptoms (r = 0.23) even accounting for maternal depression; father–child conflict mediated this link for externalizing behaviors (r = 0.57).
Ramchandani et al., 2008 [30]	N/A	Prospective population cohort.	Fathers in the UK.	10,975 fathers and their children	Paternal: - Edinburgh Postnatal Depression Scale. Child: - Development and Wellbeing Assessment.	Paternal depression gave 1.72 OR for psychiatric outcomes in children after adjustment for maternal depression and paternal education.
Van den Berg et al., 2009 [29]	N/A	Prospective population cohort.	Mothers and fathers at 20 weeks of pregnancy.	5463 fathers	Brief Symptom Inventory, depression subscale.	1.29 increased relative risk of excessive infant crying for every Standard Deviation increase in paternal depression, accounting for confounders including maternal depression.
Paulson and Bazemore, 2010 [28]	N/A	Meta-analysis.	Studies measuring peripartum paternal depression prevalence in 1980–2009.	43 studies, total n = 28,004	Number of cases of paternal depression per total study population.	Meta-estimate of paternal peripartum depression prevalence 10.4%; moderate correlation with maternal depression.
Sweeney and MacBeth, 2016 [27]	N/A	Systematic review.	Studies with measures of paternal depression and childhood (<22 years old), internalizing and/or externalizing symptoms.	21 studies met criteria	Internalizing and externalizing behaviors measured with validated tools.	Paternal depression correlates with negative childhood outcomes, mediators identified.
Hamil et al., 2021 [36]	Fathers and Babies Home Intervention.	Feasibility pilot study of intervention.	Fathers already enrolled in home visiting programs, at least 18 years old, co-parenting, and English-speaking.	30 fathers	Post-intervention surveys on feasibility and utility.	The pilot program was very well received, feasible, and acceptable, and showed evidence of effectiveness based on the initial focus groups and survey responses.
Kavanagh et al., 2021 [37]	Digital app-based parenting intervention. The app included modules on caring for babies, role changes, self-care, and supporting one’s partner.	Randomized controlled trial.	Co-parenting couples at least 18 years old, expecting their first child, English-speaking, with Internet access.	388 fathers	- Edinburgh Postnatal Depression Screening (EPDS); - Psychosocial Super Dimension Scale; - Couples Satisfaction Index; - Medical Outcomes Study; - Social Support Survey.	- High satisfaction with app; - 37% of participants logged in 2 or more times; - Average of 1–3 of 9 available modules used; - Improvement in parenting self-efficacy scores.
O’Brien et al., 2016 [38]	CBT.	Integrative review.	English articles discussing treatment of paternal perinatal depression.	12 studies	N/A	Little evidence exists examining the efficacy of CBT in paternal perinatal depression treatment.

In summary, paternal depression is prevalent, particularly rampant, and impactful in the perinatal period. This has diverse and significant consequences for both child and maternal mental health outcomes demonstrated across a host of studies. Significant mediating factors include the parental relationship, life and financial stressors, and lack of social support. Innovative programs such as those described above, incorporating partner- and family-based approaches as well as hybrid use of technology and in-person visits, hold promise for addressing the widespread and consequential occurrence of paternal depressive illness. A gap in the research exists when it comes to the direct link between effectively reducing paternal depression and the consequent improvement in children’s outcomes.

### 3.4. Parenting Interventions

Parenting and parent–child interactions significantly impact the persistence of internalizing and externalizing behaviors in childhood, and these behaviors can subsequently result in difficulties with mental health, academics, and employment [39] in adolescence and adulthood. Importantly, a parent’s mental health influences the quality of parenting and parent–child relationships. Evidence suggests that a parent with mental health concerns may be more critical, disengaged, demonstrate poor coping skills, and be less able to promote child autonomy [40]. Further, parental depression has been connected to irritability, withdrawal, and intrusiveness, which can create a system of negative and inconsistent parenting [41]. It is therefore critical to investigate whether treatments targeting parenting skills and/or the parent–child connection can lead to improvements in children’s mental health. Similarly, it is valuable to examine the impact of such interventions on parental mental health as well as how parental mental health influences the effectiveness of such interventions (Table 4). 

#### 3.4.1. Effect of Parental Depression on the Success of Parenting Interventions

One study assessed parents and their children at baseline and followed their participation in a community-based program called SNAP (Stop Now and Plan), a combined parent management training and child-focused CBT intervention [42]. The aim of the study was to assess the impact of maternal depression on the effectiveness of the program for improving children’s externalizing behaviors. This intervention resulted in a decrease in externalizing behavior in all children in the intervention group. Children of non-depressed mothers showed the greatest reduction in externalizing symptoms in the program and children of depressed mothers who remitted demonstrated the next greatest reduction. Children of depressed mothers who remained depressed continued to show a decrease in symptoms; however, these remained at the borderline-clinical level [42]. This highlights the need for the treatment of maternal depression to optimize the impact of this parenting intervention and is suggestive that this could be the case with other parenting interventions as well.

An article published In 2022 comb”ned ’ata from two clinical trials of Parent–Child Interaction Therapy (PCIT) within an ethnically diverse sample and sought to investigate the role of ameliorating parents’ mental health symptoms in improving the child’s mental health outcomes [40]. The study first demonstrated that post-intervention, there was a significant reduction in both children’s internalizing and externalizing symptoms, as well as reductions in parental depressive symptoms and parenting stress. Importantly, reductions in parenting stress after PCIT intervention mediated improvements in child internalizing and externalizing symptoms [40]. However, the data did not support reductions in parental depression as a mediator for improving child mental health outcomes as initially hypothesized. Nonetheless, as PCIT is primarily known for its effectiveness for externalizing symptoms, it is valuable to consider the impact it can have on a child’s internalizing symptoms, parental stress, and parental depression. 

#### 3.4.2. Promising Parenting Interventions for Parents with Depression

A 2009 randomized controlled trial investigated a family group-based cognitive behavioral intervention aimed to improve effective parenting as well as coping skills for the children to reduce stress related to the parent’s depression. The study consisted of a sample of 111 families with mothers or fathers with a current or past episode of major depressive disorder during their child’s lifetime [43]. The intervention led to significant reductions in children’s internalizing symptoms, as well as children’s self-report of depression, anxiety, and internalizing symptoms at 12 months post-initiation of the intervention. Additionally, parents in the intervention group had lower depressive symptoms as compared to the control group at 12 months post-intervention [43]. A follow-up study using this sample found parenting skills to be a significant mediator of the effectiveness of this intervention on children’s mental health outcomes [41]. The results suggest that improvement in parenting skills in parents with a history of depression is a vital component in reducing externalizing and depressive symptoms in their children. 

An aforementioned trial evaluated the effects of an integrated parenting intervention that combines behavioral parent training (BPT) and cognitive behavioral therapy (CBT) for mothers with at least mild depressive symptoms and their children with ADHD [14]. The authors argued that since evidence suggests that maternal cognitions, which can be influenced by depression, are connected to parenting behaviors, it follows that targeting these cognitions (with CBT) alongside targeting parental skills (with BPT) could lead to greater parenting outcomes. As such, mothers with depressive symptoms and their children with ADHD were randomized to receive the combined BPT and CBT treatment or BPT alone. The study demonstrated improvements in mother’s cognitions towards their child’s behavior [14]. Specifically, the study found that those receiving the integrated parenting intervention, as compared to BPT alone, demonstrated less negative parenting via more child-crediting attributions for behaviors post-treatment. This suggests that adding a CBT component to the standardized behavioral parenting approach can be particularly important for improving the parenting outcomes of mothers with depressive symptomatology who have children diagnosed with ADHD. 

Given that the large majority of the literature on parenting interventions’ impact on children’s externalizing and internalizing symptoms focuses on behavioral interventions, an extensive systematic review and meta-analysis focused on data for Attachment- and Emotion-Focused (AE) parenting interventions [39] which aim to focus on the parent–child relationship. The authors highlight that though behavioral parent training [BPT] is effective in particular at reducing externalizing symptoms, there remains an estimated 25–33% of children for which the intervention is not effective. Importantly, it is noted that the parent’s mental health has been demonstrated as a moderator for this outcome [39], which underscores the utility of treating parents’ psychiatric needs alongside parenting interventions. The review found that improvements in the child’s externalizing behaviors were mediated by parental sensitivity, and the improvements in internalizing behaviors were mediated by parental emotion socialization [39]. While the AE interventions did not result in improvements in parental mental health, the authors suggest that parenting interventions targeting parental mental health more specifically may be useful given that previous research supports the idea that parent training programs seem to demonstrate greater improvements in variables that are most directly related to the intervention. 

#### 3.4.3. Parenting Interventions as Prevention and Treatment for Parental Depression or Stress

The “Mom Power” study [44,45] investigated a community-based intervention for mothers and their children with reported positive outcomes for maternal mental health, parenting measures, and connection to care. While children’s mental health post-intervention was not directly measured, the studied outcomes are suggestive of this. Parenting stress significantly decreased post-intervention and the large majority (>90%) of mothers felt the intervention aided them in understanding their child’s needs and the manner in which to respond. It improved access to treatment afterwards as well, specifically in terms of self-reported improved engagement in follow-up services. 

Several articles explored parenting interventions during the prenatal, postpartum, and infancy stages. A systematic review demonstrated the positive impact of group pregnancy care [46] on mental health and wellbeing outcomes for mothers, including significant decreases in depressive symptoms. A comprehensive meta-analysis of RCTs or quasi-RCTs investigated the impact of kangaroo care, which involves continuous skin-to-skin contact with the mother, or another parent if the mother is unavailable, along with the encouragement of exclusive breastfeeding [47]. There was a high level of evidence for a reduced risk of moderate to severe postpartum maternal depressive symptoms in those in the kangaroo care group as compared to the control group. Evidence with a lower level of certainty additionally showed significantly lower maternal stress and maternal anxiety at follow-up, and significantly higher mother–infant attachment and bonding scores in the kangaroo care group versus the control condition. Based on their literature review that included three trials investigating the impact on paternal health, kangaroo care was associated with a decrease in relationship problems with spouses, positive effects in the home environment, and fathers with improved involvement in childcare, reciprocity, sensitivity, and less intrusiveness as compared to control groups [47]. Taken together, these findings emphasize the power of this intervention, given that enhancements in infant–parent bonding and interactions at a critical stage of life can have long-standing impacts on a child’s development. However, data demonstrating the child development or mental health outcomes related to kangaroo care were not included. 

One RCT of parenting interventions in the infancy period specifically studied treatment for mothers with depression along with studying child outcomes [48]. The RCT examined the impact of a group-based intervention that targeted the mother–infant relationship (HUGS) as compared to the control playgroup for mothers with mild or major depression. All mothers received CBT treatment for depression prior to randomization. No significant group differences were seen immediately following the intervention; however, at the six-month follow-up, the dyads from the HUGS treatment group demonstrated improved measures of their connection, including better affective involvement and verbalization from parents and improved bonding, as compared to the control group. There were no differences seen between groups for parenting stress and child development outcomes. Of note, mothers from each treatment group had substantial improvement in parenting stress scores post-CBT treatment, which could have decreased the likelihood of any differences emerging post-parenting intervention. Child development outcomes were assessed only six months post-intervention; longer-term follow-up in addition to an increased sample size may be needed. Additionally, the control playgroup facilitated by clinicians may have had more beneficial effects than intended [48].

Overall, interventions targeting parenting and the parent–child connection are a critical component of a child’s treatment, and the interplay between child and parental mental health is an important factor for clinicians to consider in order to optimize these interventions. The literature supports the notion that parenting interventions in parents with depression can lead to improvements in children’s mental health as well as parent–child bonding. Research suggests that maternal depression can impact the effectiveness of a parenting intervention and supports the notion that treatment for parental mental health alongside the intervention is worthwhile for maximizing children’s improvements. The evidence also supports the notion that parenting interventions alone can reduce parental depressive symptoms and parenting stress, including treatment programs during the prenatal and postpartum stages. Improvements in parenting stress and parenting skills were found as mediators of the effectiveness of parenting interventions. Additionally, it was found that adding a CBT component to the standardized behavioral parenting approach can be particularly important for improving the parenting outcomes for mothers with depression.

#### 3.4.4. Summary

Overall, interventions targeting parenting and the parent–child connection are a critical component of a child’s treatment, and the interplay between child and parental mental health is an important factor for clinicians to consider in order to optimize these interventions. The literature supports the idea that parenting interventions in parents with depression can lead to improvements in children’s mental health as well as parent–child bonding. Research suggests that maternal depression can impact the effectiveness of a parenting intervention and supports the notion that treatment for parental mental health alongside the intervention is worthwhile for maximizing children’s improvements. The evidence also supports the claim that parenting interventions alone can reduce parental depressive symptoms and parenting stress, including treatment programs during the prenatal and postpartum stages. Improvements in parenting stress and parenting skills were found as mediators of the effectiveness of parenting interventions. Additionally, it was found that adding a CBT component to the standardized behavioral parenting approach can be particularly important for improving the parenting outcomes for mothers with depression.

## 4. Discussion

This review examined the current literature in order to determine if the treatment of parental depression has an effect on childhood mental health issues. Ultimately, we found that treating depression in parents improves the parental–child bond and the overall efficacy of parenting, thus leading to improvements in childhood outcomes. Our review also revealed useful considerations in the treatment of parental depression that may in turn have positive effects on children. 

The literature suggests that childhood outcomes are most significantly improved when maternal depression is treated to remission early in its course [22,24], with medications often proving highly effective. In other words, this indicates that the earlier depression is identified and treated in parents, the better, implying that it is critical to screen and intervene early for parental depression, even when evaluating a child as the referred patient. Many studies have shown the particular importance of addressing maternal depression in the perinatal period [49]. We also found that the perinatal period is a significant risk factor for the development of depression in fathers and may prove to be a contributor to both maternal depression and depression in their children [27,28,33]. Thus, as a preventative measure in childhood mental health, this perinatal period is a critical time for intervention and support of both parents. 

Regardless of the treatment modality in addressing both maternal and paternal depression, it appears that the strongest positive outcomes for offspring were as a result of improvements in the parental–child bond [10,27]. In fact, therapeutic programs which are focused on improving the parent–child bond through relationship-building and engagement strategies may be by themselves part of the treatment for parental depression [18,20]. Ultimately, it may be argued that it is in fact the quality of the relationship and bonds within the family structure that impacts the immediate and long-term mental health outcomes of children. In order to address depression, the family must be treated and thought of as a unit.

Furthermore, our review of parenting interventions in populations of depressed parents revealed that targeting both parent depression and parenting skills in conjunction appears to be critical to improvements in child mental health outcomes. Although the studies reviewed utilized different interventions and measures, parenting interventions by themselves that did not address the depression of the parent may be less effective [42]. Specifically, interventions, such as CBT, that targeted mothers’ cognitions towards their children’s behaviors were found to be more efficacious for improving parenting outcomes than traditional BPT alone [14]. We also found evidence in the literature suggesting that mothers receiving medication for the treatment of depression saw an improvement in their parenting skills and decreased depression in their offspring [25,26]. Thus, if parenting interventions by themselves are not as effective as expected, then one may consider screening the parent for depression and offering treatment in conjunction with parenting skills interventions.

### 4.1. Limitations of the Papers Reviewed

Most of the studies reviewed focused on individual parents within a family system, primarily the mother. There were some studies that looked at the father, but these did not evaluate the effect of interventions for parental depression on childhood mental health outcomes. The studies reviewed also did not consider families outside of the model of the nuclear Western family structure. This leaves out the lived experiences of many families and, thus, limits the generalizability of these studies. There is a significant gap in community-informed research encompassing the diversity of family make-up and cultural backgrounds that influence children’s wellbeing.

### 4.2. Limitations of This Review

There are inherently some limitations to this review. We excluded papers that were not written in English and those that were not peer-reviewed. Furthermore, there may be pertinent articles found outside the databases we utilized in our search. We attempted to minimize the introduction of unintentional bias by dividing the screening of articles within our group of authors. However, as we are all based in a clinic that is centered around a family-based approach to child psychiatric care, we may have inadvertently introduced some bias for this approach within this review. Additionally, as we could not find studies directly evaluating the effect of treatment of paternal depression or caregivers outside of the nuclear family structure on childhood mental health outcomes, we cannot draw definitive conclusions about these relationships.

## 5. Conclusions

The interconnected nature of the family system is complex and multifaceted. This review highlights that the quality of the relationship with a caregiver is at the core of childhood wellbeing and that parenting stress and relationships with their offspring are at the core of parental wellbeing. Parental depression can lead to disconnection and impact the quality of this essential relationship, which may in turn contribute to both short- and long-term effects on their children’s mental health.

Importantly, our review highlights several large gaps in the literature. We noted a relatively under-researched area of paternal depression’s impact and treatment on the family system, as a vast majority of the research in the area reviewed involves mother–child dyads. More effort should be made to understand the causes of this disparity as well as novel ways to address the mental health and wellbeing of fathers. There is also a lack of information on the impact on the child of concurrently addressing depressive symptoms and stress in all their significant caregivers. Just as the child is not in isolation, the parent–child dyad is not in isolation and is impacted by myriad other interpersonal and systemic processes. The question then becomes, how do we develop programs and practices that strengthen the relationships and functioning of all those within the child’s community?

We may first start with our current understanding of the research as highlighted by this review. Across studies, one central theme emerged, of the importance of interventions that improve the parent–child relationship on mental health and behavioral outcomes of children. To accomplish this in family systems in which parents are struggling with depression, our review found that it is most efficacious to offer treatment of parental depression, either with individual therapy or psychiatric medications, in conjunction with parenting skills training. It is also particularly important to note that improvements in childhood internalizing symptoms may lag behind parental improvements in depression; in one study the gap in time of improvements seen in children after remission of maternal depression was approximately six months [20]. Several innovative programs have been developed, such as the FAB [36] and Mom Power [44,45], that may model how to deliver parenting skills and parental therapeutic interventions for depression, however, further modalities for a variety of clinical settings and family structures should be explored and developed. Given the potentially lifelong impact of parental depression on offspring, it is imperative to continue to develop programs that decrease barriers to care, such as by providing targeted time-limited treatment and/or treatment in a digital format [17,37]. Through continued research into the complicated interrelationship between parental mental health and child wellbeing, we will be able to provide more targeted, effective, and equitable treatment for the individuals and families we care for in our everyday practice.

## Figures and Tables

**Table 4 ijerph-21-00367-t004:** Effect of Parenting Interventions in Parents with Depression on Children.

	Intervention	Type of Study	Inclusion Criteria	Sample Size	Outcome Measures	Summary of Major Results
van Loon et al., 2011 [42]	SNAP (Stop Now and Plan); a 14-week combined group-based parent management training and group-based child-focused CBT intervention.	Longitudinal.	Children aged 6–12 and their mothers. Children included lived with their mothers, scored in the 98th percentile of the externalizing subscale of the CBCL, and were without developmental delay.	101 mother–child dyads	Parent: - Beck Depressive Inventory-II. Child: - Child Behavioral Checklist.	- Overall decrease in externalizing behavior in children, regardless of maternal depression status (ES 0.64, *p* < 0.01). - For depressed mothers, there was a significant decrease in depression scores post-treatment as compared to pre-treatment (ES 0.45, *p* < 0.01). - Children of non-depressed mothers showed the greatest reduction in externalizing symptoms in the program and children of depressed mothers who remitted demonstrated the next greatest reduction. - Children of mothers who had persistent depression after intervention showed the least improvement in externalizing symptoms, continuing to exhibit borderline clinical symptoms.
Compas et al., 2009 [43]	Family group-based cognitive behavioral intervention.	Randomized controlled trial.	Parents with current or past major depressive disorder during their child’s lifetime, and their children aged 9–15 years old. Multiple children within each family were included.	111 families consisting of 95 mothers, 16 fathers, and 155 children	Parent: - Beck Depression Inventory-II (BDI-II); - Structured Clinical Interview for DSM (SCID). Children: - CES-D (child self-report); - CBCL and YSR; - Schedule for Affective Disorders and Schizophrenia for School-Age Children—Present and Lifetime Version (K-SADS-PL).	Significant reductions in CES-D (d = 0.42, *p* < 0.01), YSR anxiety/depression (d = 0.5, *p* < 0.01), YSR internalizing (d = 0.57, *p* < 0.01), and parent BDI-II (d = 0.26, *p* < 0.05) at 12 months post-initiation of the intervention.
Compas et al., 2010 [41]	Family group-based cognitive behavioral intervention	Follow-up to RCT.	Parents with current or past major depressive disorder during their child’s lifetime, and their children aged 9–15 years old. If multiple children in one family were included in the original study, one child randomly selected to represent each family.	111 families consisting of 95 mothers, 16 fathers, and 111 children	Parent: - Beck Depression Inventory-II (BDI-II). Child: - Responses to Stress Questionnaire, parental depression version; - Center for Epidemiologic Studies-Depression scale (CES-D); - Child Behavior Checklist (CBCL) and Youth Self-Report (YSR). Family: - Iowa Family Interaction Rating Scales (IFIRS).	- The effect for the mediator of positive parenting was such a large proportion of the direct effect that the change in CES-D that the effect size calculated was out of range (1.04).
McCabe et al., 2022 [40]	Culturally modified versions of PCIT.	Combined data from two clinical trials of Parent–Child Interaction Therapy (PCIT).	Families included had children with clinically significant behavior problems as measured by a caregiver on the Eyberg Child Behavior Inventory (ECBI) Intensity Scale and did not have a diagnosis of autism, intellectual disability, or psychosis.	72 families with 2–7-year-old children	Parent: - Parenting Stress Index-Short Form (PSI-SF); - Beck Depression Inventory (BDI-IA). Child: - Child Behavior Checklist (CBCL); - Eyberg Child Behavior Inventory (ECBI).	- Significant reduction in both children’s internalizing (*p* < 0.001; d = 1.15) and externalizing (*p* < 0.001, d = 1.77) symptoms; - Reductions in parent depressive symptoms (*p* = 0.001, d = 0.44) and parenting stress (*p* < 0.001; d = 0.89); - Reduction in parenting stress after PCIT intervention mediated improvements in child internalizing and externalizing symptoms (*p* = 0.01); - Data did not support reductions in parental depression as a mediator for improving child mental health outcomes.
Novick et al., 2022 [14]	Integrated parenting intervention that combines behavioral parent training (BPT) and cognitive behavioral therapy (CBT).	RCT.	Mothers with at least mild depressive symptoms (10 or greater on BDI-II) and their biological children ages 6–12 who met DSM-IV criteria for ADHD.	98 mother–child dyads	Child: - Children’s Attribution Style Questionnaire-Parent Version (CASQ-P). - Expected Outcome Questionnaire. - Disruptive Behavior Disorders Checklist (DBD). Family: - Dyadic Parent–Child Interaction Coding System (3rd ed.; DPICS-III).	- Those receiving the integrated parenting intervention, as compared to BPT alone, demonstrated less negative parenting via more child-crediting attributions for behaviors post-treatment (*p* = 0.01).
Jugovac et al., 2022 [39]	Attachment- and Emotion-Focused (AE) parenting interventions.	Systematic review and meta-analysis.	AE studies that measured an externalizing and/or internalizing child outcome were included (not specific to studies of parents with depression).	43 studies (54 articles) were included in this review, including 3 quasi-experimental studies, all others were RCTs. Studies included 5542 children ages 0 to 18 years and their caregivers	- 74% of the studies included parent-report measures; however, child self-report and teacher or clinician reports were additionally included; - The majority included broad measures such as the CBCL.	- AE interventions improved the child outcomes of internalizing (small–moderate effect) and externalizing (small effect) symptoms; however, they did not find improvements in parent mental health as discussed in the text; - Improvements in externalizing and internalizing symptoms were mediated by parental sensitivity and parent emotion socialization, respectively.
Muzik et al., 2015 [44] and Rosenblum et al., 2017 [45]	Mom Power: 13-session parenting and self-care skills group program.	Pilot trial (Muzik et al., 2015) [44]. Community based RCT (Rosenblum et al., 2017) [45].	High-risk mothers (e.g., trauma exposure, poverty, mental health problems) and their young children (<6 years old).	99 mother–child pairs initially recruited (pilot study) 122 high-risk mothers and their young children (RCT)	- Maternal: Postpartum Depression Screening Scale; National Women’s Study PTSD Module. Parenting: - Parenting Stress Index—PSI; Caregiving Helplessness Questionnaire; - Connection to care: Connection to Services Tracking Sheet and 6-item self-report; - Intervention satisfaction self-report. - Children’s outcomes were not directly measured.	- For the subsample of women with reported interpersonal trauma history, there were significant decreases in depression (*p* < 0.05) and PTSD (*p* < 0.05) symptoms in the intervention group and no mental health changes for this subsample in the control group; - Parenting stress significantly decreased in the intervention group (<0.05) but not in the control group; - The large majority (>90%) of mothers felt the intervention aided them in understanding their child’s needs and the manner in which to respond.
Buultjens et al., 2021 [46]	Group pregnancy care.	Systematic review.	RCTs and observational studies studying group pregnancy care with standard pregnancy care.	Nine studies including five RCTs and four observational studies	Maternal: - Perceived Stress Scale (PSS); - Pregnancy Distress Questionnaire (PDQ) - CES-D. - Children’s outcomes were not directly measured.	- When targeted education was included in the group pregnancy care model, there were significant reductions in depressive symptoms; - For the subset of women at higher risk for psychological symptoms, there was a reported decrease in postpartum depression, stress, and anxiety symptoms.
Pathak et al., 2023 [47]	Kangaroo care, which includes continuous skin-to-skin contact with the mother and support for breastfeeding only.	Meta-analysis.	RCTs or quasi-RCTs comparing kangaroo mother care to no kangaroo care for preterm or low-birth-weight infants.	30 studies including 7719 preterm or low-birth-weight infants	Maternal mental health, mother–infant attachment and bonding, and paternal mental health outcome measures.	- High-level evidence for reduced risk of moderate–severe postpartum depressive symptoms (relative risk: 0.76); - Evidence with a lower level of certainty showed lower maternal stress and anxiety at follow-up, and higher mother–infant attachment and bonding scores; - Limited evidence that father–infant connections are improved; - No effect seen on paternal depression.
Holt et al., 2021 [48]	Four-session group-based intervention that targeted the mother–infant relationship (HUGS); both intervention and control groups followed CBT treatment for postnatal depression.	RCT.	Mothers (aged 18 years or older) diagnosed with major or minor depression using the SCID and their infants (<12 months). Mothers could not be in treatment for depression (medication or therapy).	77 mother–infant dyads	- Primary outcome measures: - Parent–Child Early Relational Assessment; - Parenting Stress Index (PSI). Child: - Ages and Stages Questionnaire; - Short Temperament Scale for Infants.	- At the six-month follow-up, the dyads from the HUGS group had improved measures of connection, including better affective involvement and verbalization from parents and improved bonding, as compared to the control group; - No differences were found in the PSI or child development outcomes (see text).

## Data Availability

No new data were created.

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
