# Peer review of "Healing Together: A Narrative Review on How Psychiatric Treatment for Parental Depression Impacts Children"

_ijerph, 2024, doi:10.3390/ijerph21030367_

Round 1

Reviewer 1 Report

Comments and Suggestions for Authors

The issue of child well-being is becoming increasingly relevant in today's complex (critical) life situations (pandemic, military conflicts, natural disasters). While the influence of the individual on large-scale social phenomena remains extremely limited, the analysis of the contribution of parental personality traits (in this case, depressive symptoms) to the well-being of their children can be very useful. This is important for the diagnosis of the development of children and family relations, for the development of scientifically based programmes for the prevention and correction of psychological problems in children, for the correction of violations in the relations between children and parents, and for the provision of professional psychological assistance to the family.

Remarks and questions to the authors:

1. You need to justify the points analysed in the Results section: psychotherapy, psychopharmacology, etc.

2. There are no clear criteria for the selection of the articles (other than that they were written in English). What was the level of acceptability and appropriateness?

3. There should be a table with the selection criteria, the type of care, the age of the parents and the children, etc.

4. Why would a reader be interested in the week in October when the search took place? How did the university librarian assist with the research?

5. Paragraph 2.4. Line 93. What was the reason for the support given to the co-authors of the article? And even if this was the case, what is the significance of this for the manuscript?

6. There should be clearer conclusions: which therapies have been more effective, what age of the children or other socio-demographic data contribute to the well-being of the children, etc.

7. It is recommended that the results are presented in tables, graphs or figures.

Author Response

Thank you for your thoughtful and thorough comments. I believe your thoughts have truly added some much needed rigor and strength to our manuscript. 

Response to your comments:

  1. To address this suggestion along with this Reviewer’s concerns expressed in item (6) we have organized all the articles into Tables based on sub-topic. This allows for more specific data about studies included to be represented graphically for purposes of understanding the points analyzed in the Results section. It is our hope that the additional details on each citation as well as the visual representation will address the Reviewer’s concern here. Additionally, each sub-section of the narrative review now includes concluding comments to help bring the data together.
  2. I have updated our eligibility criteria to include modalities of parent-focused treatment we included in our review (i.e. psychopharmacology, psychotherapy, and parenting interventions). 
  3. I appreciate this comment. We have created a table for each subsection with all categories you mention, except the selection criteria, which is highlighted in section 2. 
  4. The week of October that the search took place has been removed. Our librarian greatly assisted us in developing the search strategy in Appendix A, I would like to make sure we are giving her the credit due without adding information that is irrelevant. Please let me know if I am missing that balance. 
  5. I agree, this is superfluous and has been taken out. 
  6. As above in item(1). 
  7. As above in item(1). 

Thank you for your time, comments, and patience as we work together towards bringing what I believe is important and clinically helpful information to our colleagues. 

Best,
Michelle

Michelle Cross, DO

Reviewer 2 Report

Comments and Suggestions for Authors

1.  In general, the authors need to explain how their paper adds to the existing material from reviews of the literature and meta-analyses.

2.  Line 12.  I would recommend "or outcomes as of October 2023."

3.  Line 69.  If the goal is to assess associations between parental depression and child outcomes, it is not clear to me how assessing articles that did not include child outcomes would be useful to the goals of the paper.

4.  Line 115   I'd suggest "was" instead of "is" since the paper was published in the past.

5.  One concern I have throughout the paper is why effect sizes are reported for some past studies but not for others.  If at all possible, effect sizes should be reported for all studies included.  There are websites that will calculate effect sizes for you.

6.  Line 164.  I don't know if the underline is needed.

7.  Lines 197-198.  Did this issue have an effect of perhaps reducing apparent effects of interventions if the intervention's effects were measured at the same time as when parental depression was measured?

8.  In general, it was not clear in all studies whether the child outcomes were measured from the parent's report and/or the child's report.  This should be clearly stated for each study reviewed.

9.  Line 509.  Please expand what the "useful considerations" were.

10.  I really appreciate this paper.  If the issues can be addressed fully, it should receive special advertisement as a paper of particular interest to readers.

Comments on the Quality of English Language

I only found a couple of minor errors in English or punctuation.

Author Response

Thank you for your thoughtful and encouraging comments! Your effort has been appreciated by both myself and my team as you have undoubtedly helped us make this manuscript significantly better. 

Response to comments:

  1. We appreciate this concern and have attempted to better illustrate our points by the addition of tables which highlight key studies we reviewed in each of our subsections. Additionally, each sub-section of the manuscript now includes some sectional comments with summary statements that we believe gives it more overall cohesion and adds to our more general conclusion section.
  2. This has been corrected. Thank you for this catch.
  3. We highlight these articles in order to put into stark contrast the dearth of the literature and research in the realm of the effects of parental depression and its treatment on children/families relative to the more extensive research done with mothers in this area. We are hoping by visualizing this in the tables we have created this point will come across. However, I can appreciate that it may need to be stated more explicitly, perhaps in the conclusions. I would greatly appreciate your much more objective opinion on this. 
  4. This has been corrected, thank you. 
  5. We thank you for the suggestion of referencing the ESs, as it would add clarity and methodological rigor to the manuscript. We would be happy to present the ESs for each paper as part of Tables 1-4. However, because harvesting the ESs from each citation and transforming them to a single metric would require extensive time according to our university statistician, if needed we request additional time to do so.
  6. Thank you, this has been removed. 
  7. Perhaps this initially reduced the apparent effects of the intervention, but this was mitigated as time went on and seen as more of a lag - mother-child dyads studied here were followed up to over a year. 
  8. This is shown clearly in the newly-created tables. Thank you for this recommendation, as it helped us construct the tables. 
  9. This point was expanded in the corresponding table in this subsection.
  10. Thank you! 

Thank you again for your assistance and patience in this process. We are enthusiastic about bringing this work to a wider audience as we continue to promote the merits of a family-based approach to treatment of psychiatric conditions!

Best,

Michelle

Michelle Cross, DO

Round 2

Reviewer 1 Report

Comments and Suggestions for Authors

The authors have done a good job with the text and it has become more structured and more logical.

Some of the comments, however, have remained the same:

1. The conclusions should be clearer: which therapies were more effective, what age of the children or other socio-demographic data enhance the children's well-being, etc.

Author Response

Thank you for your additional comments. The conclusions have been expanded on (highlighted in the soon to be uploaded file). Our team believes that these changes, in conjunction with the addition of the conclusion paragraph within each subsection will provide more strength and clarity to our overall concluding thoughts and findings. Thank you again for your thoughts!